# Epidemiology of Cancers of the Small Intestine: Trends, Risk Factors, and Prevention

**DOI:** 10.3390/medsci7030046

**Published:** 2019-03-17

**Authors:** Adam Barsouk, Prashanth Rawla, Alexander Barsouk, Krishna Chaitanya Thandra

**Affiliations:** 1Hillman Cancer Center, University of Pittsburgh, Pittsburgh, PA 15232, USA; adambarsouk@comcast.net; 2Department of Medicine, Sovah Health, Martinsville, VA 24112, USA; 3Hematologist-Oncologist, Allegheny Health Network, Pittsburgh, PA 15212, USA; alexbarsouk@comcast.net; 4Department of Pulmonary and Critical Care Medicine, Sentara Virginia Beach General Hospital, Virginia Beach, VA 23454, USA; kc_thandra@yahoo.com

**Keywords:** small intestine cancer, small bowel, epidemiology, etiology, risk factors, incidence, mortality, trends, prevention

## Abstract

The latest data from the United States and Europe reveal that rare small intestine cancer is on the rise, with the number of cases having more than doubled over the past 40 years in the developed world. Mortality has grown at a slower pace, thanks to improvements in early diagnosis and treatment, as well as a shift in the etiology of neoplasms affecting the small intestine. Nevertheless, 5-year survival for small intestine adenocarcinomas has lingered at only 35%. Lifestyle in developed nations, including the rise in obesity and physical inactivity, consumption of alcohol, tobacco, and red and processed meats, and occupational exposures may be to blame for the proliferation of this rare cancer. Identification of hereditary and predisposing conditions, likely to blame for some 20% of cases, may help prevent and treat cancers of the small intestine. Studies of the neoplasm have been limited by small sample sizes due to the rarity of the disease, leaving many questions about prevention and treatment yet to be answered.

## 1. Introduction

Neoplasms of the small intestine are rare, especially considering the size of the organ. The small intestine, also known as the small bowel, makes up some 75% of the length of the alimentary canal (or digestive tract) and accounts for about 90% of its mucosal surface [1]. Nevertheless, cancers of the small intestine account for less than 5% of all gastrointestinal cancer (GI) cancer cases [1], and only about 0.6% of all cancer cases in the United States [2]. Despite their rarity, small intestine cancers are on the rise in the developed world, with an estimated growth in the incidence of over 100% in the past four decades [2].

The small intestine is located between the stomach and the large intestine and is the primary site of end absorption of nutrients from food, including proteins, lipids, and carbohydrates. It is comprised of three distinct regions: the duodenum, jejunum, and ileum. The duodenum is where most of the body’s digestive enzymes are released [3], and interestingly, it is also the most common site for cancers in the small intestine. Overall, 55–82% of small bowel neoplasms occur in this smallest region of the small intestine. Meanwhile, the jejunum accounts for 11–25% of these cancers, and the ileum for the remaining 7–17% [4,5,6,7].

Cancers of the small intestine are primarily of two etiologies: small bowel adenocarcinoma (SBA) which account for 40% of cases, and neuroendocrine tumors, which account for another 40%. Gastrointestinal stromal tumors, sarcomas, and lymphomas compose the remaining 20–25% (Table 1) [4,5,6,7]. The rarity of SBA has made it difficult to characterize the pathogenesis these two varieties fully; however, the vast differences in incidence suggests that SBA have markedly different pathogenesis, elicited by a different array of mutagens, than colon adenocarcinomas [8].

The tumor suppressor gene (TSG) *p53*, which inspects DNA integrity, has been implicated in as many as 50% of SBA cases, while the TSG adenomatous polyposis coli (*APC*) has been implicated in some 10%. Mutations in the beta-catenin pathway, essential for cellular signaling and proliferation, have been observed in another 10–40% of SBA cases. *KRAS*, an oncogene that normally functions in cellular signaling and proliferation, has been implicated in over 50% of SBA cases [9,10,11] Many, but not all, predisposing mutations (like *APC* and *KRAS*) [9], and numerous tumor suppressor and oncogenes, including p53, KRAS, and beta-catenin [9,10,11], have been implicated in the development of SBA, and newly-developed molecular therapies involving such targets have shown promise. Many, but not all, predisposing mutations (like APC [9]) and hereditary conditions (like familial adenoma polyposis [12] and Lynch syndrome [13]) are shared between small and large intestine adenocarcinomas. The hereditary nature of SBAs suggests numerous other genetic loci yet to be identified.

Among neuroendocrine tumors (NETs), multiple endocrine neoplasia type 1 is the most common predisposing condition, accounting for some 5–10% of tumors. In the condition, the multiple endocrine neoplasia type 1 (*MEN1*) gene encoding for the protein menin, a nuclear TSG that ensures accurate DNA replication, is deleted or inactivated [14].

Survival rates have markedly improved in the past decades [2], thanks to highly-sensitive means of early diagnosis. Surgically resecting the tumor remains the only known curative treatment, particularly for early-stage disease [15]. The efficacy of adjuvant chemotherapy among those getting surgical resection remains controversial. Meanwhile, little clinical data are available for the recommended treatment of advanced SBA, which, after decades of improvement, now has a 5-year survival rate of 42% [2]. Quite fittingly with the anatomy, this survival rate is above that of the large intestine, but below that of stomach cancers.

Due to its rarity, small intestine cancer has been seldom studied. Nevertheless, as its incidence has been steadily increasing, it has become imperative that we learn to diagnose better, treat, and prevent this atypical neoplasm.

## 2. Epidemiology

### 2.1. Incidence

In 2018, there were an estimated 10,470 small intestine cancer cases in the United States. This constituted an incidence rate of 2.3/100,000 people. These cases also comprised only 0.6% of all cancer diagnoses in the United States that year [2]. In contrast, there were about 145,000 cases of large intestine cancer in the United States that same year. According to 2015 data, the lifetime risk of developing small intestine cancer in the United States is about 0.3% [2], some 2–5 times less than that for cancer of the large intestine [16]. While colorectal cancer ranks as the fourth most incident in the United States, small intestine cancer comes in 23rd place [2]. Men are more increasingly likely than women to fall ill, with an incidence of 2.6 in 2018 relative to 2.0 for women. African Americans are disproportionately affected, with incidences of 4.2 and 3.5 among men and women, respectively, while Native Americans and Asians were the least likely to be diagnosed with the disease. In the United States, the median age for the first diagnosis was 66 years [2].

The United Kingdom has a comparable rate of small intestine cancer diagnoses, with an incidence of 3.1/100,000 among men and 2.2/100,000 among women. Their overall incidence the same as that of the United States: 2.6/100,000. Interestingly, males in Scotland have a higher incidence than the U.K. average (3.5/100,000), while females have a lower incidence (1.6/100,000) on [17].

According to 1998–2002 data, small intestine cancers were most common in North America and Western Europe and least common in Asia. In those years, the incidence was 1.4 and 1.0, for men and women respectively, in the United States and Sweden, while 0.7 and 0.4 in Japan [18].

### 2.2. Mortality

Mortality from neoplasms of the small intestine is 0.5/100,000 for men and 0.3 for women in the United States. The corresponding mortality rates for African Americans are 0.7 and 0.5, respectively. The median age of death is 72 years old. An estimated 1450 Americans died from small intestine cancer in 2018, which constituted 0.2% of all cancer deaths. While incidence has steadily risen, the average mortality has remained at 0.4 since 1976 [2].

Meanwhile, mortality was over twice as high in the United Kingdom, with rates of 1.0 and 0.7 for men and women respectively. Since the 1970s, small intestine cancer mortality had increased in the United Kingdom by 37%. Mortality was highest among both men and women in Scotland, and lowest in Wales for men and Northern Ireland for women. The most common age for small intestine cancer death was 85–89 years old, almost a decade greater than the United Kingdom life expectancy [17].

### 2.3. Survival

The 5-year survival for neoplasms of the small intestine in the United States is 67.6%. Survival is highly variable based on the stage at diagnosis. Among localized cases, which make up 32% of all diagnoses, 5-year survival is 85%. For cases that have spread to regional lymph nodes (35% of cases), 5-year survival is 74.6%. Meanwhile, for advanced cases with distant metastases (which comprise 27% of diagnoses), 5-year survival drops down to 42.1% [2] (Figure 1).

In England, 5-year survival is markedly lower than that of the United States, or the European average, for that matter. The 5-year survival among men is 37% in England, below the European average of 47%. Wales (41%), Scotland (44%), and Ireland (41%) are closer to the European average. The 5-year survival for women in England (36%) has an even greater discrepancy with the European average of 50%. Interestingly, women in Scotland and Wales have worse survival than those in England (33% and 34%, respectively), unlike their male counterparts [17].

Among men, Switzerland has the highest small intestine cancer 5-year survival rate in Europe at 61%, which is still about 7% lower than the average survival for men and women in the United States. Among women, the highest survival is in Belgium, also at 61%. The lowest performing country for men in Europe is Latvia, with a 20% survival rate, and Malta for women, with a 31% rate. Poland reported an overall survival rate of 42%, below the European average, but higher than that of England and several other Western European nations [19].

Survival is highly variable based on the type of small intestine tumor. Neuroendocrine tumors (NETs) have a 5-year survival of around 85%, while small bowel adenocarcinomas (SBA) show a survival of only about 35% in the United States. The median overall survival for NETs is 9.3 years. The other two common types, lymphomas, and sarcomas, both have 5-year survivals of around 60–70% [4].

### 2.4. Trends

The incidence of small intestine cancer has been increasing in the developed world over at least the past half-century. In 1975, the incidence in the United States was 1.1, relative to 2.4 in 2018. This constitutes an increase of 118% over the past 43 years. Meanwhile, mortality has remained constant at 0.4 since 1976. In turn, survival rates have improved in par with increasing incidence (Figure 2). The 5-year survival was 33.1% in 1975, compared to 67.6% in 2015. This constitutes a doubling in average survival over a span of 40 years [2].

In the United Kingdom, mortality has increased over the past decades, though not as much as incidence, indicating an improvement in survival as well. In 1971, small intestine mortality was 0.6, relative to 0.9 today, an increase of 37%. Meanwhile, in half that time (since 1993), small intestine incidence in the United Kingdom has grown about 132%, more than in the United States [17].

In the United States, the frequencies of the types of small intestine cancer have changed over the past decades. From 1985 to 2005, the proportion of SBA dropped from 42% to 33%. The proportion of neuroendocrine tumors grew from 28% to 44% in this same time period, overtaking adenocarcinomas as the primary tumor type. This increase may be reflective of better diagnostic tools for identifying and classifying NETs. An increase in NETs (which have a better prognosis relative to SBA) may partially explain improvements in survival in the United States. Meanwhile, from 1985 to 2005, the proportions of sarcomas and lymphomas remained stable at around 17% and 7%, respectively [4,20].

## 3. Etiology and Risk Factors

### 3.1. Etiology

Adenocarcinomas arise as the proliferation of the mucosal epithelial cells lining the small intestine. As in the large intestine, such benign polyps can become immortalized and ultimately transform into adenocarcinomas after a latent period of 10–20 years. Adenomas, or polyps of granular cells that release mucus, are the precursors of adenocarcinoma [21]. In the large intestine, only about 10% of all adenomas progress to adenocarcinoma [22]. Due to the rarity of small intestine cancers, and the lack of regular screening procedure comparable to the colonoscopy, it is unknown whether the same proportion applies. Although the small intestine comprises 90% of the mucosal surface of the digestive tract, small bowel neoplasms account for less than 5% of all GI cancers, indicating a marked difference in pathogenesis [1]. Theories for this difference include the shorter transit time through the small intestine, which limits exposure to dietary carcinogens, as well as the lower density of microbiota which may convert bile salts into carcinogenic deoxycholic acid [23]. Adenocarcinomas of the small intestine are most common in the duodenum, where bile and pancreatic juices are released to aid in digestion [4,24].

By contrast, neuroendocrine tumors, also classified as carcinoid tumors, are most common in the ileum [4,24]. They are defined as epithelial tumors with primarily neuroendocrine differentiation. NETs are rare and slow-growing, and while some of their features are shared among all, most of their features are unique to their organ of origins [25]. These tumors arise from neuroendocrine cells, such as enterochromaffin cells, scattered throughout the small intestine (and the entire GI tract). As their name suggests, these cells produce hormones and cytokines to regulate digestive system function [26]. NETs have been found to be genomically stable tumors, meaning they acquire few genetic mutations over the process of carcinogenesis [27]; however, studies suggest that to compensate, many display dysregulation of microRNAs [28].

Sarcomas, i.e., cancers of the smooth muscle and connective tissue lining the intestines, as well as lymphomas, comprise the remaining 20–25% of small intestine neoplasms.

### 3.2. Non-Modifiable Risk Factors

#### 3.2.1. Race and Ethnicity

In the United States, African Americans are more likely to fall ill with small intestine cancer by 1.6–1.8 times relative to white Americans. Asians, Pacific Islanders, Native Americans, and Alaskans are the least likely to fall sick with the disease. African Americans also have proportionally higher mortality [2].

#### 3.2.2. Age

The median age for small intestine cancer diagnosis in the United States is 66 [2]. The median age for diagnosis is comparable in the United Kingdom; however, the most common age for incidence in the United Kingdom was 80–84 [17]. The median age for small intestine mortality is 72 in the United States [2], while the “peak” age for mortality is 85–89 in the United Kingdom, over a decade greater than the average life expectancy [17]. After around 40 years of age, the risk of small intestine cancer begins to increase, and it does not appear to level out until around 90 years old [17].

#### 3.2.3. Sex

Males are more likely to be diagnosed with, and die from, small intestine cancer than females. In the United States, the gender discrepancy in incidence is about 1.3:1 and in mortality about 1.6:1, suggesting lower survival rates among men [2]. Interestingly, in the United Kingdom, while men are still more likely to be diagnosed, survival rates for men are actually higher than those for women [17]. Differences in diet, carcinogen exposure, and metabolic rate, among others, may underlie the sex difference in small intestine bowel incidence and mortality.

#### 3.2.4. Hereditary Mutations

Those with familial adenomatous polyposis (FAP) have a germline *APC* mutation that predisposes towards the growth of adenoma polyps. Patients with FAP have thousands of polyps growing in their intestinal lining by the age of 10–12. Over time, the risk of these adenomas transforming to adenocarcinoma grows exponentially, and by age 40, virtually all patients with FAP will have received a colorectal cancer diagnosis [29]. The small intestine is the second most common site for adenocarcinoma among those with FAP. In a study of 1255 patients, about 5% had been diagnosed with small bowel adenocarcinoma (SBA). Half of those cancers were found in the duodenum [30]. The risk of small intestine adenocarcinoma among those with FAP is 330 times higher than for the general population [12,31]. In fact, duodenal adenocarcinoma was the primary cause of death among FAP patients who had undergone a coloproctectomy [32,33]. Hereditary mutations associated with SBA and NETs have been summarized in Table 2.

##### Lynch Syndrome

Lynch syndrome is a hereditary defect in mismatch repair which is the primary risk factor for non-polyposis colorectal carcinoma (CRC). Those with the condition have a 20% of developing CRC by age 50 years and a 50% chance of developing it by age 70 years [34]. The relative risk for SBA ranges from about 25 in the early stages of Lynch syndrome to up to 291 for those with an *MLH1* mutation [13,35,36]. Nevertheless, the lifetime risk for SBA among Lynch syndrome patients remains low, at around 1% according to a French registry [34,37]. While it is not currently recommended to screen Lynch syndrome patients for SBA, it has been advised to screen SBA patients for Lynch syndrome, considering the syndrome could predispose to more common, undetected cancers around the body [38].

##### Peutz–Jeghers Syndrome

Peutz–Jeghers syndrome (PJS) is a rare disorder which occurs due to an autosomal dominant defect in the STK11 suppressor that predisposes to hamartomatous polyps in the GI tract. The relative risk for those with PJS is 520 times that of the general population [39].

##### Multiple Endocrine Neoplasia Syndrome Type 1

Multiple endocrine neoplasia syndrome type 1 is an autosomal dominant defect in the *MEN1* gene that greatly predisposes to NET of the upper GI tract as well as tumors of the pituitary, thyroid, and pancreas. Defects in *MEN1* account for 5–10% of all GI NETs [40].

##### Neurofibromatosis Type 1

Neurofibromatosis Type 1 is an autosomal dominant defect in the *NF-1* that predisposes to NETs as well as sarcomas. With an incidence of 1/4000, it is the most common cancer-predisposing hereditary condition. Patients with neurofibromatosis type 1 have a 2–4 times increased risk of developing neuroendocrine and stromal tumors around the body [41].

#### 3.2.5. Inflammatory Bowel Disease

Inflammatory bowel disease (IBD) is most often caused by Crohn’s disease or ulcerative colitis. Crohn’s disease is a hereditary autoimmune disorder that causes inflammation throughout the intestine, affecting the ileum most commonly. This inflammation culminates in the abnormal release of growth cytokines, excess blood flow, metabolic free radicals, and other factors that predispose towards carcinogenesis [8,42]. The relative risk for SBA among those with Crohn’s disease ranges from 17 to 41. Unlike sporadic SBA, these cases arise more often at younger ages and particularly in the ileum, where the most inflammation occurs [43]. The cumulative risk is estimated at 0.2% after 10 years of Crohn’s disease, but 2.2% after 25 years [44]. The SEER study suggested a lifetime risk of 1.6% of SBA among those with Crohn’s disease, almost 3 times that of the average American [45].

#### 3.2.6. Celiac Disease

Celiac disease is an autoimmune disorder triggered by the consumption of gluten, prevalent in nearly 1% of the population. The autoimmune response culminates in digestive tract inflammation and ultimate destruction of enterocytes, as well as premalignant changes that have been shown to increase the risk of SBA as well as small intestine lymphoma. In a small study of 235 celiac disease patients, 8% were found to have SBA, which suggests a relative risk of about 13 [46]. A Swedish study pegged the relative risk at 10 [47]. A British study of 395 cases of small intestine cancer found celiac disease was implicated in 13% of SBA cases (mostly in the jejunum) and 39% of lymphomas [48].

A prospective French study with SBA patients found that predisposing or hereditary diseases were implicated in 20% of cases: 8.6% had Crohn’s disease, 3% had FAP, 3% had Lynch syndrome, 1.5% had celiac disease, and 0.8% had PJS. Among metastatic patients, 14.7% had Crohn’s disease, and 5.9% had Lynch syndrome [49,50].

### 3.3. Modifiable Risk Factors

#### 3.3.1. Diet

Diets that have a high content of animal fat and protein have been associated with a higher risk of small intestine cancer, with correlation coefficients of 0.61 and 0.75, respectively [51]. A separate case–control study found a 2–3 fold increased risk of cancer among those with high red and salt-cured/smoked meat intake [52]. Another case–control study found an increased risk of cancer among males (but not females) who consumed high amounts of heterocyclic aromatic amines, found in smoked and fried meats [53]. A different study disputed the carcinogenic effects of meat specifically but found an association between small intestine cancer and saturated fats, found in high amounts in meats [54]. Red and processed meat have likewise been implicated in colorectal cancer (with respective relative risks of 1.12 and 1.15) [55]. Processed meat was designated as a carcinogen, and red meat as a probable carcinogen, in 2015 by the International Agency for Research on Cancer (IARC), due to their likely effect on the development of cancers of the small and large intestines. Meanwhile, fibers from grains, vegetables, and beans have been found to decrease the risk of cancers of the small and large intestines, likely by promoting faster transit times and thus limiting carcinogen exposure [56].

#### 3.3.2. Alcohol

Several studies have found a slight positive association between alcohol and SBA, while several others have not [52,57,58]. One European study split the difference, finding that while red wine did not increase SBA incidence, consumption of beer and spirits did [59]. Heavy alcohol consumption had been found to increase the risk of CRC by 20–40% [60], leading many to assume that it has a similar effect on SBA, even if small sample sizes due to the rarity of the disease preclude definitive results.

#### 3.3.3. Smoking

Several studies have found that cigarette smoking increases the risk of both SBA and NET [61,62,63], while one European study only found a significant effect on NET [59,62]. As with alcohol, the effect of smoking on CRC is more definitive, with a relative risk of 1.18, leading the IARC to definitively conclude that tobacco is a cause of CRC [64].

#### 3.3.4. Obesity

Numerous studies have investigated a correlation between obesity and small intestine cancer, with mixed results. One study found a relative risk of 2.8 among those diagnosed with obesity [65]. Another bizarrely placed the relative risk at 1.44 among the overweight and 1.1 among the obese [66]. Other studies reported similarly confounding results, such as an increased risk among obese men, but not women [67], or an increased risk among white obese veterans, but not black [68]. Several studies found no effect, or even an inverse effect, of body mass index (BMI) on small intestine cancer [52,57,58]. All these studies were plagued by small sample sizes due to the rarity of the condition, though looking at data on CRC seems to reaffirm the carcinogenic effect of obesity and lack of physical activity, at least on the latter portion of the intestines.

#### 3.3.5. Biliary Tract Diseases

A Danish study found an elevated risk of small intestine cancer among those suffering from gallstones. The cancers were mostly of the NET variety [69]. Consuming a diet rich in fibers and avoiding obesity has been proven to reduce one’s risk of developing gallstones. Another study confirmed the findings, though concluded the increase in diagnosis of NET might be due to enhanced medical surveillance for the gallstones [62]. A competing theory invokes the idea that bile can indeed be carcinogenic, and those with gallstones often produce more bile than normal. However, yet a different study found that those who underwent a cholecystectomy, or the removal of the gallbladder, also saw an increased risk of small intestine cancers [63].

#### 3.3.6. Occupational Hazards

Several studies have found numerous occupations that were associated with an increased risk of small intestine cancer, though the significance of the findings is challenged by the rarity of the disease. One study found an elevated risk among Australian Nuclear Workers, despite the fact that radiation exposure for said workers is tightly regulated [70]. Other occupations, implicated in a separate study, included shoemakers, structural metal preparers, construction painters and other construction workers, bookkeepers, machine fitters, and welders, as well as people with regular occupational exposure to organic solvents and rust-preventive paint containing lead [62].

#### 3.3.7. Vitamins and Medications

One study which intended to uncover an association between ultraviolet-B radiation and small intestine cancer, in fact, contradicted its own hypothesis, finding those at northern latitudes were most at risk, perhaps due to vitamin D deficiency [71]. Interestingly, vitamin D has in fact been shown to reduce CRC [72]. However, long-term consumption of certain medications, such as nonsteroidal anti-inflammatory drugs (NSAIDs) and corticosteroids, which seem to lower the risk of CRC [73], have not been found to have the same effect on small intestine cancer [63].

## 4. Prevention

A steady increase in the incidence of small intestine neoplasms in the developed world suggests that environmental components are triggering the onset of the disease. Hereditary or predisposing conditions have been implicated in only 20% of small intestine cancers. Meanwhile, numerous studies have suggested diet, smoking, alcohol, obesity, and occupational hazards in the development of small intestine cancers. Although research is limited, studies of CRC, a close cousin of SBAs, suggest significant improvements in cancer reduction among those who practice prevention via behavioral modification.

In the last four decades, obesity in the United States has more than doubled from 15% in 1979 to 38% today [74]. This trend has coincided with, and perhaps even fueled, a doubling in the cases of small intestine cancer. Fortunately, many Americans today are also making improvements in their lifestyle. From 2006 to 2012, the proportion of Americans meeting physical activity guidelines increased from 41% to 50% [75]. Regular smoking has been reduced from around 40% of U.S. adults to 15% in the past four decades as well [76].

Physical activity, weight loss, smoking, and alcohol cessation, avoidance of red, smoked and processed meats, and consumption of fibers have all been shown to reduce small intestine cancer risk to varying degrees, as well as reduce the much more common CRC significantly. Avoidance of occupation-related mutagens and radiation is also likely to decrease one’s risk of numerous cancers, including that of the small intestine. While vitamin D and medications like steroids and NSAIDs have not been found to have a definitive effect on small intestine cancers, they do seem to have a protective role against CRC. Small sample sizes constrain studies into the risk factors and prevention of small intestine cancer and thus may overlook potentially preventive agents and behaviors due to a lack of statistical significance.

## 5. Conclusions

SBA is a rare neoplasm, though its incidence has more than doubled in the last few decades in the developed world. Fortunately, mortality has not kept pace with incidence, thanks to improved prevention and treatment, as well as an increase in the proportion of neuroendocrine tumors (which have a better prognosis) relative to adenocarcinomas. The increase in incidence may be partially due to better surveillance and diagnostics, but may also have been fueled by lifestyles, such as the rise in obesity, smoking, and consumption of alcohol and red and processed meats. Some 20% of cases are likely due to underlying hereditary or predisposing conditions. Survival for adenocarcinomas remains low, and prevention by behavioral modification could save thousands of lives each year. However, studies of risk factors and treatments for small intestine cancers have been limited by the rarity of the disease, and the best course of action is preventing and treating the deadly and elusive neoplasm.

## Figures and Tables

**Figure 1 medsci-07-00046-f001:**
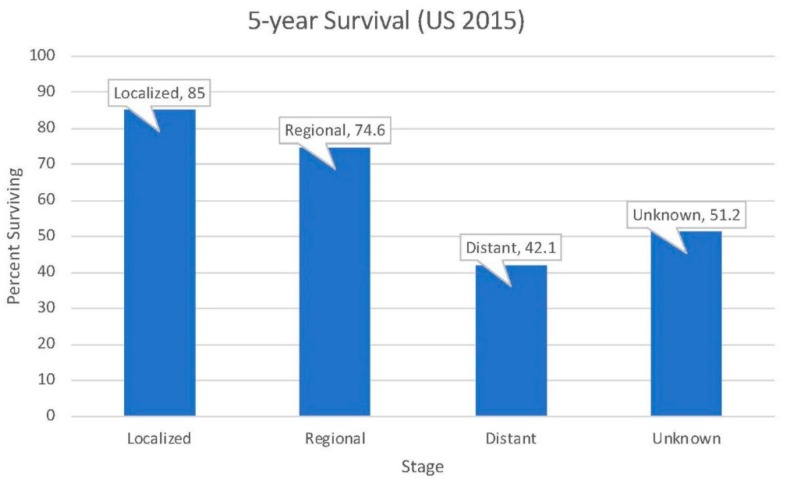
Five-year survival for neoplasms of the small intestine in the United States (data obtained from the Surveillance, Epidemiology, and End Results (SEER) database).

**Figure 2 medsci-07-00046-f002:**
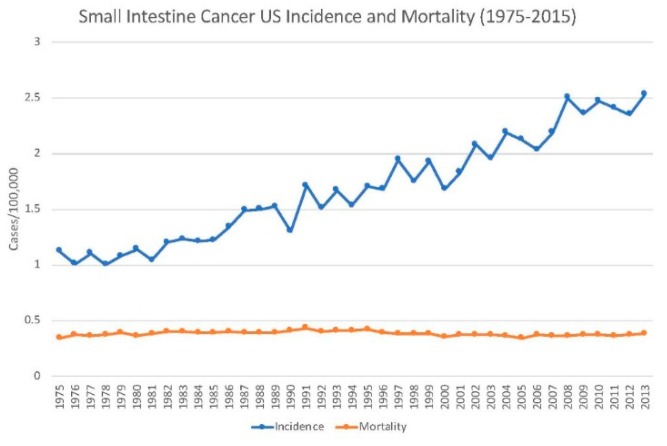
Incidence and mortality rates of small intestine cancer in the United States from 1975 to 2015 (data obtained from the Surveillance, Epidemiology, and End Results (SEER) database).

**Table 1 medsci-07-00046-t001:** Small intestinal cancer etiologies and incidence.

Etiology	Adenocarcinoma	Neuroendocrine (i.e., Carcinoid)	Lymphoma	Sarcoma (i.e., Stromal)
Proportion	30–40%	35–42%	~17%	~8%
U.S. incidence (per 100,000)	0.7	1.0	0.4	0.2
Most probable location	Duodenum	Ileum	Throughout GI tract	Throughout GI tract

**Table 2 medsci-07-00046-t002:** Hereditary mutations associated with small bowel adenocarcinoma (SBA), neuroendocrine tumors (NETs), and increased risk of cancers.

Predisposing Condition for SBA	Genetic Mutation	Increased Risk	Predisposing Condition for NET	Genetic Mutation	Increased Risk
Familial adenomatous polyposis (FAP)	*APC* deletion/inactivation	~**330**×	Multiple endocrine neoplasia type I	Defective *MEN1*	Accounts for 5–10% of NET in GI tract
Lynch syndrome (HNPCC [Hereditary nonpolyposis colorectal cancer])	Mismatch repair-deficient	**25**–**291**×	Neurofibromatosis 1 (also predisposes to sarcoma)	Defective *NF1*	2–4× increased risk of all tumors
Peutz–Jeghers syndrome (PJS)	*STK11* suppressor defect	**520**× (both NET and SBA)

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
