# Peer review of "Epidemiology of Cancers of the Small Intestine: Trends, Risk Factors, and Prevention"

_medsci, 2019, doi:10.3390/medsci7030046_

Round 1

Reviewer 1 Report

The study by Barsouk et al reviewed the recent data on epidemiology of cancers of the small intestine. The cancers of the small intestine are rare. The study is informative and will bring attention to recent trends of increase of the incidence. It is possible that better diagnostic imaging technology contributes the increase of incidence. However, the paper does not have a section on disease symptoms and diagnosis. 

Author Response

The study by Barsouk et al reviewed the recent data on epidemiology of cancers of the small intestine. The cancers of the small intestine are rare. The study is informative and will bring attention to recent trends of increase of the incidence. It is possible that better diagnostic imaging technology contributes the increase of incidence. However, the paper does not have a section on disease symptoms and diagnosis.

-

Thank you for the valuable suggestions and feedback. We believe that diagnosis, signs, and symptoms are a separate topic from epidemiology. In this review paper, we have mainly focused on epidemiology and risk factors. Writing on diagnosis, signs and symptoms will make the topic too big and will deviate from the main purpose of this review. We wanted to concentrate only on Epidemiology, trends, risk factors and preventive strategies. We are considering writing it as a separate review paper in the near future.

Reviewer 2 Report

This paper studied the epidemiology of the small intestine (trends, risk factors, and prevention).

This paper is interesting.

I would suggest to replace “2.3   Mortality” with “2.3   5-year survival”.

Author Response

This paper studied the epidemiology of the small intestine (trends, risk factors, and prevention).

This paper is interesting.

I would suggest to replace “2.3   Mortality” with “2.3   5-year survival”.

- Thank you for the valuable suggestions and feedback. We have made the appropriate changes as suggested. We have replaced 2.3 Mortality with 2.3 Survival.

- Thank you for the valuable suggestions and feedback. We have made the appropriate changes as suggested above. I hope that we have answered and made all the changes to the manuscript as suggested and to your satisfaction.

Reviewer 3 Report

This is a particularly well written paper on the epidemiology review paper on cancers of the small intestine. The topic is well covered, and the major issues are appropriately analyzed and discussed.

I suggest some changes to improves the impact of this interesting paper:

1-   It could be particularly useful for the reader and for the clarity of the manuscript it would be very useful to add a Table reporting the cellular classification of small intestine cancers, with their incidence.

2-   A Table reporting the major molecular abnormalities observed in adenocarcinomas and neuroendocrine tumors would be very useful (seerSchrorck et al, JAMA 2017; Bonck et al, J Clin Invest 2013; Francis et al, Nat Genet 2013; Karppathakis et al, Clin Cancer Res 2016).

3-   A brief mention of the major biochemical pathways altered in adenocarcinomas and neuroendocrine tumors of the small intestine would be important (in the introduction).

4-   Data of mortality related to the various cellular subtypes would complete the data already analyzed in the manuscript.

Author Response

This is a particularly well written paper on the epidemiology review paper on cancers of the small intestine. The topic is well covered, and the major issues are appropriately analyzed and discussed.

- Thank you

I suggest some changes to improves the impact of this interesting paper:

1-   It could be particularly useful for the reader and for the clarity of the manuscript it would be very useful to add a Table reporting the cellular classification of small intestine cancers, with their incidence.

- Thank you for the valuable suggestion. We have added a table (Table 1) with cellular classifications and their incidence in Introduction, Paragraph 3.

2-   A Table reporting the major molecular abnormalities observed in adenocarcinomas and neuroendocrine tumors would be very useful (seerSchrorck et al, JAMA 2017; Bonck et al, J Clin Invest 2013; Francis et al, Nat Genet 2013; Karppathakis et al, Clin Cancer Res 2016).

- Thank you for the valuable suggestion. We have added a table (Table 2) which shows the various genetic abnormailities observed in adenocarcinomas and neuroendocrine tumors in Hereditary Mutations, Paragraph 1.

3-   A brief mention of the major biochemical pathways altered in adenocarcinomas and neuroendocrine tumors of the small intestine would be important (in the introduction).

- Thank you for the suggestion. We have discussed in brief about the major biochemical pathways altered in adenocarcinomas and neuroendocrine tumors of the small intestine in Introduction section, paragraph 4 and 6.

4-   Data of mortality related to the various cellular subtypes would complete the data already analyzed in the manuscript.

- thank you for the suggestion. We searched the literature but were not able to find the mortality breakdown by the various cellular types. If you can suggest the source we can include the data in the review. Thank you.

- Thank you for the valuable suggestions and feedback. We have made the appropriate changes as suggested above. I hope that we have answered and made all the changes to the manuscript as suggested and to your satisfaction.